# Fibroin Delays Chilling Injury of Postharvest Banana Fruit via Enhanced Antioxidant Capability during Cold Storage

**DOI:** 10.3390/metabo9070152

**Published:** 2019-07-23

**Authors:** Juan Liu, Fengjun Li, Lei Liang, Yueming Jiang, Junjia Chen

**Affiliations:** 1Guangdong Key Lab of Sugarcane Improvement & Biorefinery, Guangdong Provincial Bioengineering Institute (Guangzhou Sugarcane Industry Research Institute), Guangdong Academy of Sciences, Guangzhou 510316, China; 2Key Laboratory of Plant Resources Conservation and Sustainable Utilization, Guangdong Provincial Key Laboratory of Applied Botany, South China Botanical Garden, Chinese Academy of Sciences, Guangzhou 510650, China

**Keywords:** banana fruit, peel browning, ROS, antioxidant capability, fibroin

## Abstract

storage Banana fruit after harvest is susceptible to chilling injury, which is featured by peel browning during cold, and it easily loses its nutrition and economic values. This study investigated the role of fibroin treatment in delaying peel browning in association with the antioxidant capability of postharvest banana fruit during cold storage. Compared to the control fruit, fibroin-treated fruit contained higher amounts of Pro and Cys during overall storage as well as higher glutathione (GSH) during the middle of storage. Conversely, fibroin-treated fruit exhibited a lower peel browning index and reactive oxygen species (ROS) level during overall storage as well as lower contents of hexadecanoic acid and octadecanoic acid by the end of storage compared to control fruit. In addition, fibroin-treated banana fruit showed higher activities of superoxide dismutase (SOD) and ascorbate peroxidase (APX) in relation to upregulation *SOD*, *CAT,* and *GR* as well as peroxiredoxins (*MT3* and *GRX*) during the middle of storage. These results highlighted the role of fibroin treatment in reducing peel browning by enhancing the antioxidant capability of harvested banana fruit during cold storage.

## 1. Introduction

Banana (*Musa acuminata* Colla cv. Cavendish) is a globally consumed fruit with attractive taste and appealing commercial value, and it is widely distributed in tropical and subtropical regions. However, banana fruit is vulnerable in low temperatures, which brings about chilling injury symptoms once it is stored at a temperature <13 °C [1]. These chilling injury symptoms in postharvest banana fruit can result in undesirable texture and unsuitable marketability.

Peel browning is one of the typical symptoms of chilling injury in postharvest banana fruit during cold storage. It has been suggested that peel browning could result from the contact of phenolics with polyphenol oxidase (PPO) and/or peroxidase (POD), which can be due to the loss of cell membrane integrity caused by the degradation of cellular membrane lipids [2,3]. With regard to membrane integrity, reactive oxygen species (ROS) is involved more particularly in damaging the membrane structure by oxidizing biomolecules such as proteins and lipids [4]. Therefore, it could be speculated that peel browning of banana fruit during cold storage could be closely associated with cellular ROS level and redox equilibrium.

It is documented that oxidative stress resulting from overproduction of ROS has been related to banana fruit quality during cold storage [1]. Accumulated evidence indicates that endogenous nonenzymatic and enzymatic antioxidants participate in eliminating ROS to minimize the destructive effect on plants [5,6,7]. Among them, primary metabolites such as amino acids, organic acids, and sugars as well as lipids may play critical roles in maintaining the physiological activity of plants under cold stress [8,9,10]. In particular, glutathione (GSH) and ascorbic acid (AsA) as well as phenols and carotenoids, based on their reducing action with H_2_O_2_, lipid peroxides, and oxidized proteins, are essential for protecting plants from being damaged by oxidative stress [11,12]. In addition, ROS-utilizing enzymes including POD, superoxide dismutase (SOD), catalase (CAT), and ascorbate peroxidase (APX) mainly account for catalyzing the conversion to H_2_O_2_ or removing H_2_O_2_ directly. It is worth mentioning that metallothionein (MT), glutaredoxin (GRX), and thioredoxin (TRX) are indispensable for plants to display reductase activities based on their protein cysteinyl residues [13]. Furthermore, a series of genes encoding antioxidant enzymes are triggered to change at transcription levels, and then the antioxidant ability is enhanced, and the repair capability of peroxiredoxins is stimulated in response to cold stress [14,15].

Fibroin is a hydrophilic natural polymer protein derived from the silkworm, and it shows good potential to serve as an edible coating of postharvest fruit and vegetables due to its safety, mechanical properties, and flexible gas and water diffusivity. Fibroin has proved to be effective in not only delaying the senescence and ripening of bananas and strawberries [16] but also alleviating chilling injury of banana fruit stored at low temperature [17]. Our previous work revealed that fibroin treatment increased the cold tolerance of banana fruit by maintaining ATP levels during storage [17]. Currently, no studies have been reported on the role of fibroin treatment in protecting banana fruit against oxidative stress during cold storage. Therefore, to elucidate the action of fibroin, the effects of fibroin treatment on amino acids, primary metabolites, and antioxidants in relation to the gene expressions of the antioxidant enzymes and peroxiredoxins were investigated in this work.

## 2. Results 

### 2.1. Changes in the Peel Browning Index of Harvested Banana Fruit during Cold Storage 

As shown in Figure 1, the peel browning index of harvested banana fruit increased continuously from stage I (0 d) to stage III (4 d) during cold storage. In this study, 0.1 g L^−1^ fibroin solution was used to treat banana fruit for 3 min and then stored for 4 d at 6 °C. For the control fruit group, the peel browning index increased rapidly from 0 to 2.35, followed by a slow increase, which indicated that the banana peel was almost fully browned. For the fibroin-treated banana fruit, the peel browning index exhibited a slower increase from 0 to 1.55 and from 1.55 to 2.95. In contrast to the control fruit, fibroin treatment delayed significantly (p < 0.05) the peel browning of harvested banana fruit during cold storage.

### 2.2. Changes in Amino Acid Contents of Harvested Banana Fruit during Cold Storage

Table 1 shows the contents of free amino acids after fibroin treatment in peels of harvested banana fruit. The peel of banana fruit contained higher contents of Asn, Asp, His, and Glu, followed by Ser, Ala, and Gly, but it had lower contents of Cys and Pro. Nonprotein amino acids included ornithine (Orn), γ-aminobutyric acid (GABA), phosphatidylserine (P-Ser), and phosphatidyl ethanolamine (P-Eta). It is worthy to note that P-Ser and P-Eta were closely related to plant membrane structure. Some amino acids showed decreases in content, while some decreased and then increased during cold storage. However, the Pro content increased as storage time extended. Furthermore, fibroin treatment increased the Pro content. Interestingly, no Cys was detected in control fruit, but Cys content tended to increase after fibroin treatment. In addition, fibroin treatment enhanced the contents of most amino acids at corresponding storage times in contrast to the control fruit. 

### 2.3. Changes in Primary Metabolites of Harvested Banana Fruit during Cold Storage

The contents of primary metabolites in peels of harvested banana fruit are shown in Table 2. Twenty-five metabolites including amino acids, sugars, organic acids, lipids, alcohols, and others were detected. The contents of six amino acids decreased, except Gly and Val, as storage time extended. Fibroin treatment decreased the contents of seven amino acids during the middle of storage but increased the contents of all the amino acids by the end of storage. The contents of glucopyranose, 2-keto-d-gluconic acid, and d-glucuronic acid decreased during storage, but fibroin treatment increased sugar contents remarkably, except α-d-glucopyranoside, by the end of storage. The contents of organic acids decreased generally. Fibroin treatment increased the contents of citric acid, malic acid, ethanedioic acid, and 6-hydroxy-2-aminohexanoic acid significantly by the end of storage. The levels of hexadecanoic acid and octadecanoic acid decreased and then increased during storage; fibroin treatment decreased them significantly by the end of storage in contrast to the control. These data suggest some primary metabolites could be involved in delaying chilling injury, which might be positively influenced by fibroin treatment.

### 2.4. Changes in Antioxidants, Antioxidant Enzyme Activities, and ROS Levels in Peels of Harvested Banana Fruit during Cold Storage

As illustrated in Figure 2, banana peels contained high contents of ASA and total phenols but low contents of GSH and total carotenoids. With increasing storage extension, the contents of ASA and total phenols were reduced, whereas the contents of GSH and total carotenoids were elevated. By contrast, fibroin treatment increased the contents of GSH during the middle of storage but decreased the total carotenoids by the end of storage. As shown in Figure 3, POD and APX activities increased, but SOD and CAT activities increased and then decreased during storage. However, fibroin treatment enhanced the APX activity during the middle of storage and increased the SOD activity during the entire storage time, which might benefit the redox status in banana peels. Meanwhile, fibroin treatment inhibited the production of ROS (Figure 4). On the other hand, higher levels of antioxidants as well as enhanced activities of antioxidant enzymes would help to eliminate the accumulated ROS. Results demonstrated that fibroin treatment maintained better banana quality with lower ROS levels during cold storage via enhanced antioxidant contents and antioxidant enzyme activities. 

### 2.5. Changes in Expression Levels of Genes Related to Inhibition Mechanisms of Oxidative Damage and Peroxiredoxins in Peels of Harvested Banana Fruit during Cold Storage

To illustrate the influence of fibroin treatment on inhibiting the oxidative damage in peel browning of banana fruit, the expressions of several related genes, including *POD*, *SOD*, *CAT*, *GPX*, *APX,* and *GR,* and peroxiredoxins were investigated. As shown in Figure 5, except for *POD*, the relative expression levels of *SOD, CAT, GPX, APX,* and *GR* were downregulated and then upregulated as storage time extended. Fibroin treatment induced upregulation of *SOD, CAT,* and *GR* expressions during the middle of storage but downregulated the relative expression levels of *SOD*, *CAT*, *GPX,* and *GR* by the end of storage. Figure 6 presents the expression levels of *MT*, *GRX,* and *TRX*. Except for *GRX*, the relative expression levels of *MT1*, *MT3,* and *TRX* were downregulated initially and then upregulated as storage time extended. However, fibroin treatment upregulated the relative expression levels of *MT3* during storage and upregulated the relative expression levels of *GRX* by the end of storage. These data suggest that the related genes in relation to inhibition of oxidative damage could be induced by fibroin treatment, which might stimulate the upregulation of most genes during the middle of storage and, thus, might contribute to inhibiting the oxidative damage in peel browning of banana fruit during cold storage.

## 3. Discussion

### 3.1. Fibroin Treatment Reduced Chilling Injury of Banana Fruit during Cold Storage 

Chilling injury, that mainly occurs in tropical and subtropical fruit, is the main quality problem for bananas stored at low temperature [1]. To a certain extent, the physiological response to low temperature can be reflected by contents of free amino acids and primary metabolites. Previous studies conducted in citrus fruit indicated that contents of amino acids changed significantly in response to temperature stress [8]. In particular, Gln and Orn were cold-responsive to activate amino acid metabolism as a result of low temperature. Moreover, the accumulation of Gln and Orn in *Arabidopsis* showed a higher tolerance to salt and drought stresses [18]. Hence, Gln and Orn might be critical indicators to sense environmental stimuli and act as signaling compounds. On the other hand, Pro plays an important role for plants in response to abiotic stress, and it accumulates as an antioxidant to protect cellular structures [9], which was in accordance with increased Pro content in this work (Table 1). Correlation analyses indicated that the peel browning index was positively correlated with the Pro content in control fruit and fibroin-treated fruit, with *y* = 1.301*x* − 0.048 (*r* = 0.908) and *y* = 0.396*x* − 0.080 (*r* = 0.982), respectively. It could be inferred that Pro was a key factor in banana peel browning during cold storage. Interestingly, Pro has been linked with several signaling pathways, such as the ABA signaling pathway, in protecting against oxidative membrane damage [10]. It is also worthy to note that Cys was highly involved in plant resistance in response to stress as an antioxidant [19]. In this work, the content of Cys displayed a positive correlation with the peel browning index in fibroin-treated banana fruit (*y* = 0.686*x* + 0.038, *r* = 0.998, *P* < 0.05). These results illustrated that fibroin treatment could effectively delay the chilling injury of banana fruit during cold storage by increasing Pro and Cys contents. 

Organic acids and sugars contribute greatly to fruit quality, such as with taste and aroma, which could be distinctly affected by storage temperatures. The possible link between sugars and fruit quality at low temperature was reinforced by the accumulation of soluble sugars, such as glucose, sucrose, and fructose, which would enhance the chilling tolerance of banana fruit [17]. In this work, fibroin treatment increased the sugar content by the end of storage, which was beneficial in the quality maintenance of banana fruit. It was shown that contents of citric acid and malic acid gradually decreased during storage [20], but other studies showed that their contents decreased significantly at 4 °C for 90 d [8], which could be attributed to activations of aconitase and malate dehydrogenase. In addition, fibroin treatment enhanced the contents of citric acid and malic acid by the end of storage, which verified that fibroin treatment improved the quality of banana fruit. Recent investigations in pepper provided evidence about involvement of lipids in plant tolerance to cold stress [9]. The unsaturation degree of membrane lipids increased to enhance the cold tolerance [21]. Conversely, the proportion of saturated fatty acid decreased, which was in accordance with the decreased contents of two saturated fatty acids (hexadecanoic acid and octadecanoic acid) in fibroin-treated fruit compared to the control fruit by the end of storage in our work (Table 2). According to the correlation analysis, the peel browning index of fibroin-treated banana fruit was negatively correlated with the content of hexadecanoic acid (*y* = −23.67*x* + 92.01, *r* = 0.900) or octadecanoic acid (*y* = 16.01*x* + 56.38, *r* = 0.896). All these data highlighted the crucial role of fibroin treatment in delaying peel browning and maintaining quality of banana fruit via regulation of primary metabolites during cold storage.

### 3.2. Fibroin Treatment Alleviated Oxidative Damage on Banana Fruit during Cold Storage 

ROS is the main byproduct in the process of electron transport in the respiratory chain. It is known well that ROS accumulates promptly when plants are exposed to biotic and abiotic stresses. In addition, fatty acid oxidation, the photorespiratory glycolate oxidase reaction, and the enzymatic reactions of flavin oxidases are also the primary metabolic pathways responsible for ROS generation [22]. In recent years, it was shown that the accumulation of ROS could destroy the cell structure and accelerate aging [23,24], while other studies exhibited that ROS regulated metabolic process and signaling pathways [7,25]. However, whether ROS is harmful or beneficial to plants depends largely on the ROS strength in plant tissues.

As illustrated in Figure 4, banana fruit at harvest contained low levels of ROS, but the ROS accumulated quickly with increasing storage time. According to the correlation analysis, the peel browning index was positively correlated with the level of O_2_^−.^ production rate (*y* = 0.141*x* + 0.027, *r* = 0.951), H_2_O_2_ (*y* = 0.114*x* + 0.042, *r* = 0.921), or ^–^ OH^.^ (*y* = 0.140*x* +0.065, *r* = 0.879). These data exhibited that fibroin treatment could inhibit the peel browning of banana fruit by reducing the accumulation of ROS. Recent investigations in *Arabidopsis* provided evidence about involvements of malate and ROS production as well as programmed cell death in plant and animal systems [26]. Furthermore, ROS might form an important signaling network together with other signaling pathways, such as ABA signaling and Ca^2 +^ signaling under different stress conditions [27]. Among them, ROS might induce oxidative stress under abiotic stress [28]. In this work, fibroin treatment reduced ROS levels and, therefore, alleviated oxidative damage and delayed peel browning of banana fruit during cold storage. It was also in agreement with recent investigations in mangos, which revealed that cold temperatures induced oxidative stress and evoked antioxidant system defenses against ROS accumulation [29].

### 3.3. Fibroin Treatment Improved Antioxidant Ability of Banana Fruit during Cold Storage 

According to current studies, oxidative damage on plants under cold stress is attributed to overproduction of ROS [7]. However, endogenous enzymatic and nonenzymatic antioxidants in plants can eliminate ROS to maintain redox equilibrium [30]. Among these nonenzymatic antioxidants, GSH, ASA, phenols, and carotenoids constitute the ROS detoxification system and then protect the macromolecules from being oxidized by ROS [12]. The correlation analysis showed that the levels of O_2_^−.^ and − OH^.^ in fibroin-treated banana fruit were positively correlated with the total carotenoid contents, with *y* = 13.924*x* + 0.089 (*r* = 0.994) and *y* = 13.553*x* − 0.057 (*r* = 0.997), respectively. In addition, the levels of O_2_^−.^ and − OH^.^ were positively correlated with activities of POD (*y* = 2146.5*x* + 0.217, *r* = 0.975 and *y* = 2111*x* + 0.034, *r* = 0.999, respectively) and SOD (*y* = 4183.5*x* − 0.046, *r*=0.991 and *y* = 3977.5*x* − 0.106, *r* = 0.949, respectively) in fibroin-treated banana fruit. Furthermore, the levels of O_2_^−.^ and H_2_O_2_ were also positively correlated with APX activity (*y* = 186.69*x* + 0.107, *r* = 0.901 and *y* = 236.02*x* + 0.078, *r* = 0.965, respectively) in fibroin-treated banana fruit. These data indicate that the increased levels of total carotenoids and enhanced activities of antioxidant enzymes played key roles in eliminating ROS. As shown in Figure 2, higher GSH content in fibroin-treated banana fruit during the middle of storage compared to the control fruit helped to improve the antioxidant ability of banana fruit. According to the correlation analysis, the peel browning index was negatively correlated with the AsA content in control fruit (*y* = −0.109*x* + 9.167, *r* = 0.849) and fibroin-treated fruit (*y* = −0.110 *x* + 9.1007, *r* = 0.932). Moreover, the reduced peel browning index was positively correlated with the increased content of total carotenoids in fibroin-treated banana fruit in contrast to the control fruit (*y* = 1.93 *x* + 0.048, *r* = 0.911). These data suggest that fibroin treatment could delay peel browning of banana fruit by affecting AsA and total carotenoid contents. However, there was no significant difference between control and fibroin-treated fruit in relation to AsA and total carotenoids in this study. Lutein, a-carotene, and β-carotene are the main components of total carotenoids and contribute to the yellow color of banana fruit [31]. It is suggested that low temperature and respiration metabolism exert impacts on carotenoid synthesis and conversion [32]. 

POD, SOD, APX, and CAT have been long recognized as key antioxidant enzymes to remove ROS, while SOD mainly is responsible for catalyzing superoxide radicals to H_2_O_2_, and POD, APX, and CAT are involved in erasing H_2_O_2_ [33]_._ In our work, the activities of POD and APX were triggered to increase, while SOD and CAT were induced to increase initially and decrease by the end of storage. According to the correlation analysis, the decreased peel browning index was positively correlated with the increased activity of POD (*y* = 292.14x + 0.074, *r* = 0.862), SOD (*y* = 603.37*x* + 0.009, *r* = 0.984), or APX (*y* = 28.337*x* + 0.005, *r* = 0.991). These antioxidant enzymes could act as mediators of cell redox homeostasis by regulating the formation of ROS in delaying peel browning of banana fruit stored at low temperature. 

### 3.4. Fibroin Treatment Stimulated Repair Capability of Peroxiredoxins of Banana Fruit during Cold Storage 

Peroxiredoxins are involved particularly in the cellular protection against oxidative damage resulting from ROS overproduction under cold stress [34]. The relevant regulations of genes encoding enzymes against oxidative damage exerted an important influence on antioxidant defense systems and resulted in a great change in antioxidant capability. Compared to the control, fibroin treatment upregulated the expressions of *POD*, *CAT,* and *GR* during the middle of storage but downregulated the expressions of *POD*, GPX, *CAT,* and *GR* by the end of storage. In particular, it is worth mentioning that GPX catalyzes the conversion of harmful peroxides into water or alcohols [13]. Previous studies have found that GPX regulates redox status under abiotic stresses such as cold, drought, and oxidative stresses [35,36,37]. Investigations in rice provided evidence about upregulation of *GPX* genes under cold stress [36]. Moreover, *OsGPX5* plays a critical role in interaction between redox homeostasis and ER stress for rice to balance normal development and stress responses [38].

Metallothionein (MT) is mainly involved in cellular protection processes in response to various stresses [39]. It was shown that the *MT* gene was promptly upregulated under oxidative stress [39]. Other studies showed that several signaling pathways formed a network, which resulted in the regulation of stress-responsive genes including the *MT* gene [40]. As illustrated in Figure 6, fibroin treatment upregulated the transcription level of *MT3* during the entire storage time, which highlighted the crucial role of *MT* in protecting against oxidative stress induced by low temperature. 

GRX and TRX as sensor molecules are involved in recognizing oxidative stress, and they maintain the cellular redox status through protein cysteinyl residues. Plant GRX and TRX have been highlighted to participate in protective responses to oxidative stress [41]. Particularly, TRX may recognize oxidized proteins or H_2_O_2_ directly [42]. As shown in Figure 6, fibroin treatment significantly upregulated the transcription level of *GRX* by the end of storage. It is proposed that fibroin treatment could improve redox homeostasis via the regulations of *GRX* and *TRX* at the transcription level.

Taken collectively, the upregulated transcription levels of *SOD*, *CAT,* and *GR* during the middle of storage and peroxiredoxins (*MT3* and *GRX*) in banana fruit during overall storage could enhance protection against oxidative stress and, thus, could benefit in inducing the antioxidant ability of banana fruit against peel browning during cold storage.

## 4. Materials and Methods

### 4.1. Plant Materials and Treatments

Green banana fruit (*Musa acuminata* Colla cv. Cavendish) with 80% maturity (approximately 110 d after anthesis) were harvested from an orchard in Guangzhou City, Guangdong province, China. Fruit were screened for similar sizes, soaked in 0.1% Sportak solution for 3 min, and then immersed in 1 g L^−1^ fibroin solution. Fibroin (molecular weight: 200–4000 Da) was chosen from Fushengde Biological Engineering Co., Ltd., Jiangsu Province, China, according to our previous report [17]. The control and fibroin-treated fruit were air-dried, packed by 0.03 mm polyethylene bags with twenty fruit per bag, and stored at 6 °C. We took fruit at 0, 2, and 4 d as storage I, II, and III, respectively. Clear distinctions could be observed between the fibroin-treated and control fruit during the three storage periods. Fruit peels were sampled and preserved at −80 °C by liquid nitrogen before analyzing the following parameters. 

### 4.2. Evaluation of Peel Browning Index

Fifty fruit were used to evaluate the peel browning index based on the method of Jiang et al. [1]. Peel browning index was assessed by estimating the total browning area in peels of 50 individual fruits using the following visual appearance scale: 0, no browning; 1, 0%–25% browning area; 2, 25%–50% browning area; 3, 50%–75% browning area; and 4, 75%–100% browning area. The peel browning index was calculated as ∑ (browning scale × proportion of fruit at each scale).

### 4.3. Extraction and Analyses of Amino Acid Content 

Amino acid contents were analyzed according to the method in [43]. Briefly, 1 g of peel tissues was homogenized under liquid nitrogen and then extracted with 50 mL 0.01 M HCl. After centrifugation at 8000× *g*, 2 mL of the upper phase was taken and mixed with 8 mL ethanol by vortexing for 30 min. After centrifugation at 8000× *g* for 10 min, the upper phase was taken, dried by a vacuum concentrator, and re-dissolved with 1 mL 0.01 M HCl. The solution was filtered through a 0.22 µm Millipore membrane filter prior to HPLC analysis. The results of all amino acid contents were expressed as mg kg^−1^ of fruit fresh weight.

### 4.4. Extraction and Analyses of Primary Metabolites 

Determination of primary metabolites was carried out according to the literature [44]. Peel tissues (0.2 g) were homogenized under liquid nitrogen and extracted with 1.8 mL methanol. To the mixture, 200 µL ribitol was added; it was vortexed for 2 min, followed by ultrasonic extraction for 15 min at 4p, and then extracted by water-bath heating at 70 °C for 15 min. After centrifugation at 5000× *g* for 15 min at 4 °C, 100 µL of the upper phase was taken and dried by a vacuum concentration. The dried samples were derivatized with 80 µL O-methylhydroxylamine hydrochloride dissolved in pyridine, and then they were incubated at 37 °C in a vacuum drying oven. The dried samples were derivatized with 80 µL of N-methyl-N-[trimethylsily] trifluoroacetamide for 30 min at 37 °C. The mixture was filtered through a 0.22 µm Millipore membrane filter prior to GC-MS analysis. One microliter of derivatized sample was subjected to GC-MS equipped with a DB-5MS column (Agilent Technologies, 30 mm × 0.25 mm × 0.25 µm). The injector temperature was 230 °C. High-purity nitrogen gas (99.999%) was the carrier gas with a velocity 1.2 mL min^−1^. The temperature program of GC included an initial temperature of 100 °C for 1 min; a ramp of 3 °C min^−1^ to 184 °C, followed by another ramp of 0.5 °C min^−1^ to 190 °C, and kept at this temperature for 1 min; and an additional ramp of 15 °C min^−1^ to 280 °C and kept at this temperature for 5 min. The split ratio was 100:1. The content of each compound was identified by the peak area ratio of each compound to ribitol multiplied by the concentration of ribitol. The results of all primary metabolites were expressed as µg ribitol g^−1^ of fruit fresh weight (FW). 

### 4.5. Measurement and Analyses of Glutathione (GSH), Ascorbic Acid (AsA), and Total Phenols

GSH content was measured as follows. Two grams of peel tissues was homogenized under liquid nitrogen, and then 5 mL enzyme extracting solution of 0.05 mol L^−1^ Tris-HCl, 0.1 mmol L^−1^ EDTA, and 1% PVP at pH 7.6 was added. The mixture was ground in an ice bath for 10 min. After centrifugation at 15,000× *g* for 30 min at 4 °C, 2 mL of the supernatant was taken, and 1.0 mL distilled water was added. Then, 0.1 mL of 0.1 mmol L^−1^ DTNB was added to perform a reaction for 10 min at room temperature. The absorbance was measured at 412 nm. As for assay of total phenols, 5 g of fruit peel was homogenized under liquid nitrogen and then extracted with 1% HCl-methanol solution at room temperature for 2 h. After the solution was filtered, the filtrate was used to determine the content of total phenols. The filtrate (50 μL) was taken, and 3 mL distilled water was added. After the solution was mixed, 0.1 mL of 50 mmol L^−1^ FeCl_3_ and 0.1 mL of 8 mmol L^−1^ K_3_Fe(CN)_6_ were added successively and mixed again by vortexing for 3 min. The mixture was placed at room temperature for 30 min, and the absorbance was measured at 720 nm. Gallic acid was used as the standard to calculate the total phenol content. The content of AsA was measured according to a method previously reported [45]. The contents of total phenols were expressed as mg per g of fresh weight, while the contents of GSH and AsA were expressed as µg g^−1^ of fresh weight.

### 4.6. Determination of Total Carotenoids

One gram of peel tissues was homogenized under liquid nitrogen and then extracted with 2 mL of 80% acetone by vortexing for 2 min, and it was finally kept at room temperature for 1 min. After centrifugation at 10,000 rpm for 10 min, the supernatants were collected, and the absorbances of the acetone extracts were measured at 663, 646, or 470 nm using a UV-Vis spectrophotometer. The content (µg g^−1^) of total carotenoids was calculated using the following equation:


Carotenoids = [1000 × A_470_ − 3.27 × (12.21 × A_663_ − 2.81 × A_646_) − 104 × (20.13 × A_646_ − 5.03 × A_663_)]/229.


### 4.7. Analyses of Enzyme Activities

Four grams of peel tissues was powdered under liquid nitrogen and then added to 20 mL of 0.2 mol L^−1^ phosphate buffer (pH 6.8), which contained 100 μmol L^−1^ EDTA and 0.4 g PVP, and, finally, was ground in an ice bath. After centrifugation at 15,000× *g* for 15 min at 4 °C, the supernatant was taken to measure enzyme activities. The supernatant (0.05 mL) was taken, and a reaction solution containing 0.1 mL of 4.0% guaiacol, 0.1 mL of 0.46% hydrogen peroxide, and 2.75 mL of 0.5 mol L^−1^ phosphate buffer was added. The absorbance was measured at 470 nm. One unit (U) of POD was defined as 0.01 of absorbance fluctuation per minute. For assays of SOD activity, 0.05 mL of the mentioned supernatant was taken and added to a reaction solution containing 1.3 µmol L^−1^ riboflavin, 13 mmol L^−1^ methyl ammonium sulfate, 63 µmol L^−1^ NBT, 100 µmol L^−1^ EDTA, and 0.5 mol L^−1^ phosphate buffer (pH 7.8). The mixture was performed at a light intensity of 4000 lx, and the absorbance was then measured at 560 nm. One unit (U) of SOD was defined as every 50% inhibition of the reduction of NBT. For assays of CAT activity, 0.05 mL of the mentioned supernatant was taken and added to 3 mL reaction solution containing 2.95 mL of 15 mmol L^−1^ H_2_O_2_ and 0.5 mL of 0.05 mol L^−1^ phosphate buffer (pH 7.8). After the solution was mixed, the absorbance was measured at 240 nm. One unit (U) of CAT was defined as 0.001 of absorbance fluctuation per minute. For assays of APX activity, 0.1 mL of the mentioned supernatant was taken and added to 3 mL reaction solution containing 0.5 mmol L^−1^ ASA, 0.1 mmol L^−1^ EDTA, and 0.05 mol L^−1^ phosphate buffer (pH 7.0). The reaction started once H_2_O_2_ was added, and then the change of absorbance at 290 nm per minute was measured. One unit (U) of APX was defined as the amount of enzyme needed to oxidize 1 µmol ASA.

### 4.8. Measurement of ROS

The ROS level was determined according to the literature [46]. Four grams of peel tissues was homogenized under liquid nitrogen, and it was added to 20 mL of 0.065 mol L^−1^ phosphate buffer (pH 7.8) and 0.4 g PVP. The solution was filtered through four-layer gauze, the filtrate was centrifuged at 5000× *g* for 10 min, and 1 mL of the supernatant was taken and added to 0.9 mL phosphate buffer and 0.1 mL of 10 mmol L^−1^ hydroxylammoniumchlroride for 30 min at 25 °C. The mixture was taken, and 1 mL of 0.017 mol L^−1^ p-aminobenzene sulfonic acid and 1 mL of 7 mmol L^−1^ 1-naphthylamine successively were added. The reaction was performed for 20 min. Then n-butanol with the same volume was added and mixed, and then the n-butanol phase was taken after centrifugation at 5000× *g* for 10 min. The absorbance was measured at 560 nm. For H_2_O_2_ content, 5 g of peel tissue was mixed with 25 mL cold acetone and centrifuged at 6000× *g* for 15 min. The supernatant (1 mL) was taken, and 0.1 mL of 20% TiCl_4_ and 0.2 mL of 2 mol L^−1^ ammonia were added to perform the reaction. The solution was centrifuged at 6000× *g* for 15 min, the precipitate was collected and dissolved with 3 mL of 1 mol L^−1^ H_2_SO_4_, and then it was centrifuged at 5000× *g* for 20 min. The absorbance of the supernatant was measured at 410 nm. The hydroxyl radical was measured according to the method in [47]. H_2_O_2_ content was expressed as nmol kg^−1^ on a fresh weight basis, while −OH^.^ level was expressed as U mL^−1^. One unit (U) of −OH^.^ level was defined as every increase of 1 mmol L^−1^ of H_2_O_2_ per mg on a fresh weight basis. The results of O_2_^−.^ production rates were expressed as nmol kg^−1^ s^−1^.

### 4.9. RNA Isolation and Real-Time PCR

Total RNA was extracted from banana peels according to the method described by Kuang et al. [48]. RNA integrity and purity were electrophoretically verified by checking that the A260/A280 absorption ratio was between 1.9 and 2.1. PrimeScript^TM^ RT Master Mix (TAKARA-RR036A, Dalian, China) was used to synthesize the first strand cDNA by following the manufactures’ instructions. Primer premier 6.0 software (Premier, Canada) was used to design the specific primer pairs of the selected genes, which are listed in Appendix A. The specific procedure for quantitative real-time PCR was the same as our previous report [17], and the relative expression levels of target genes were determined by the equation 2^ΔΔCT^. Three biological replicates were used for this analysis.

### 4.10. Statistical Analysis

The results were expressed as means ± standard errors (SE). Significant differences between the fibroin-treated banana fruit and control fruit were examined using the software SPSS 22 (IBM, New York) via ANOVA (analysis of variance). The level of significant difference between two groups was set at *P* < 0.05. 

## 5. Conclusions 

In conclusion, this work demonstrated that fibroin treatment could increase the contents of Pro and Cys and reduce the contents of hexadecanoic acid and octadecanoic acid in peels of banana fruit. Fibroin treatment can also maintain higher GSH during the middle of storage and reduce ROS accumulation in banana peels. Furthermore, the increased activities of antioxidant enzymes (SOD and APX), upregulation of *SOD, CAT, GR,* and *MT3* expressions during the middle of storage, and upregulation of *GRX* and *MT3* expressions by the end of storage were observed in peels of banana fruit. Therefore, fibroin treatment improved the antioxidant ability of banana peels and, thus, delayed peel browning of harvested banana fruit during cold storage. The possible mechanism of fibroin involved in inhibiting peel browning of harvested banana fruit via the enhanced antioxidant defense system by fibroin treatment is presented in Figure 7. Our results proved that fibroin exhibited the potential for reducing peel browning of harvested banana fruit during cold storage.

## Figures and Tables

**Figure 1 metabolites-09-00152-f001:**
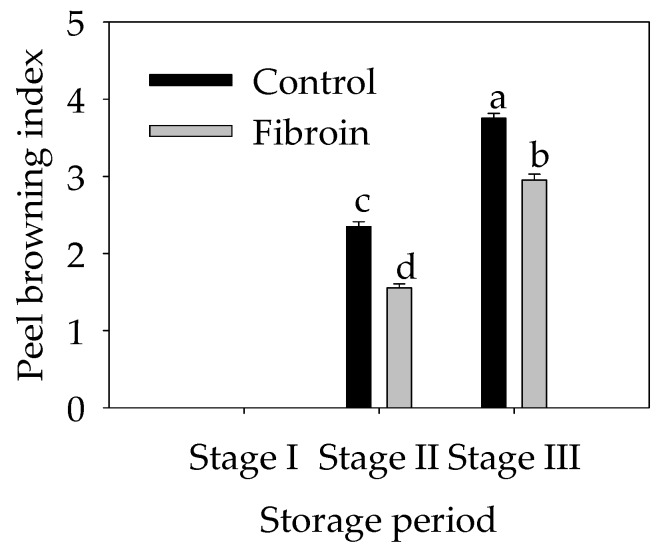
Effect of fibroin treatment on the peel browning index of harvested banana fruit during cold storage. Data are presented as means ± standard errors (*n =* 3). The means with different letters are significantly different (*p* < 0.05).

**Figure 2 metabolites-09-00152-f002:**
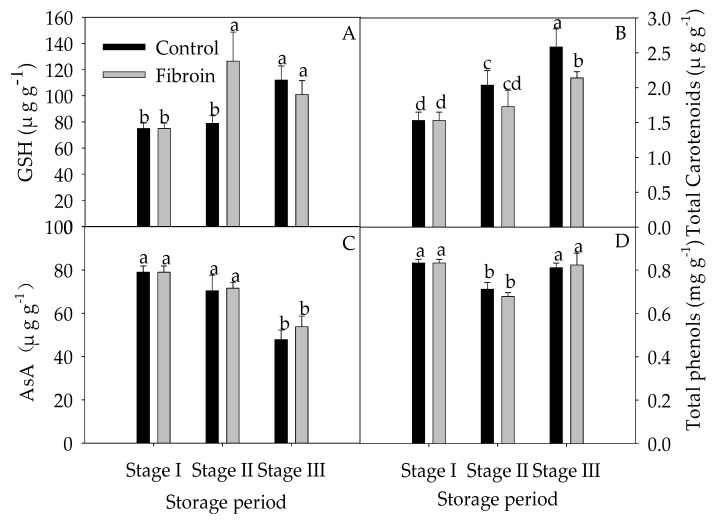
Effect of fibroin treatment on contents of glutathione (**A**), total carotenoids (**B**), ascorbic acid (**C**), and total phenols (**D**) in peels of harvested banana fruit during cold storage. The data are presented as means ± standard errors (*n* = 3). The means with different letters are significantly different (*p* < 0.05). GSH, glutathione; AsA, Ascorbic Acid.

**Figure 3 metabolites-09-00152-f003:**
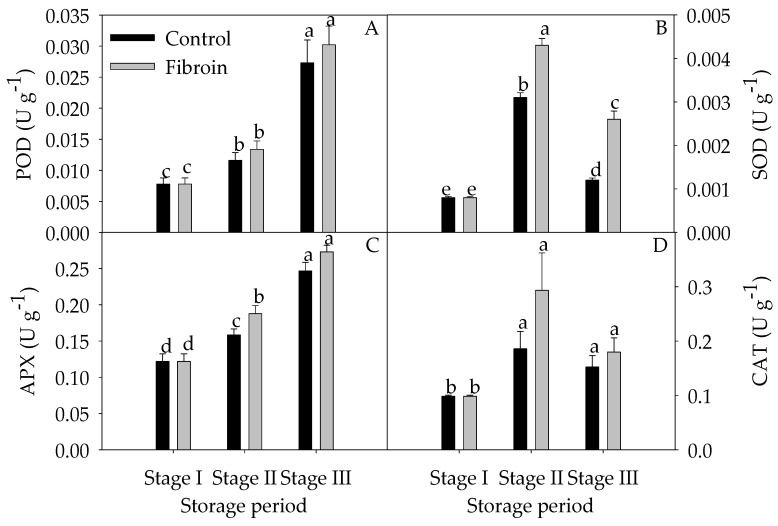
Effects of fibroin treatment on activities of peroxidase (POD) (**A**), superoxide dismutase (SOD) (**B**), ascorbate peroxidase (APX) (**C**), and catalase (CAT) (**D**) in peels of harvested banana fruit during cold storage. The data are presented as means ± standard errors (*n =* 3). The means with different letters are significantly different (*p* < 0.05).

**Figure 4 metabolites-09-00152-f004:**
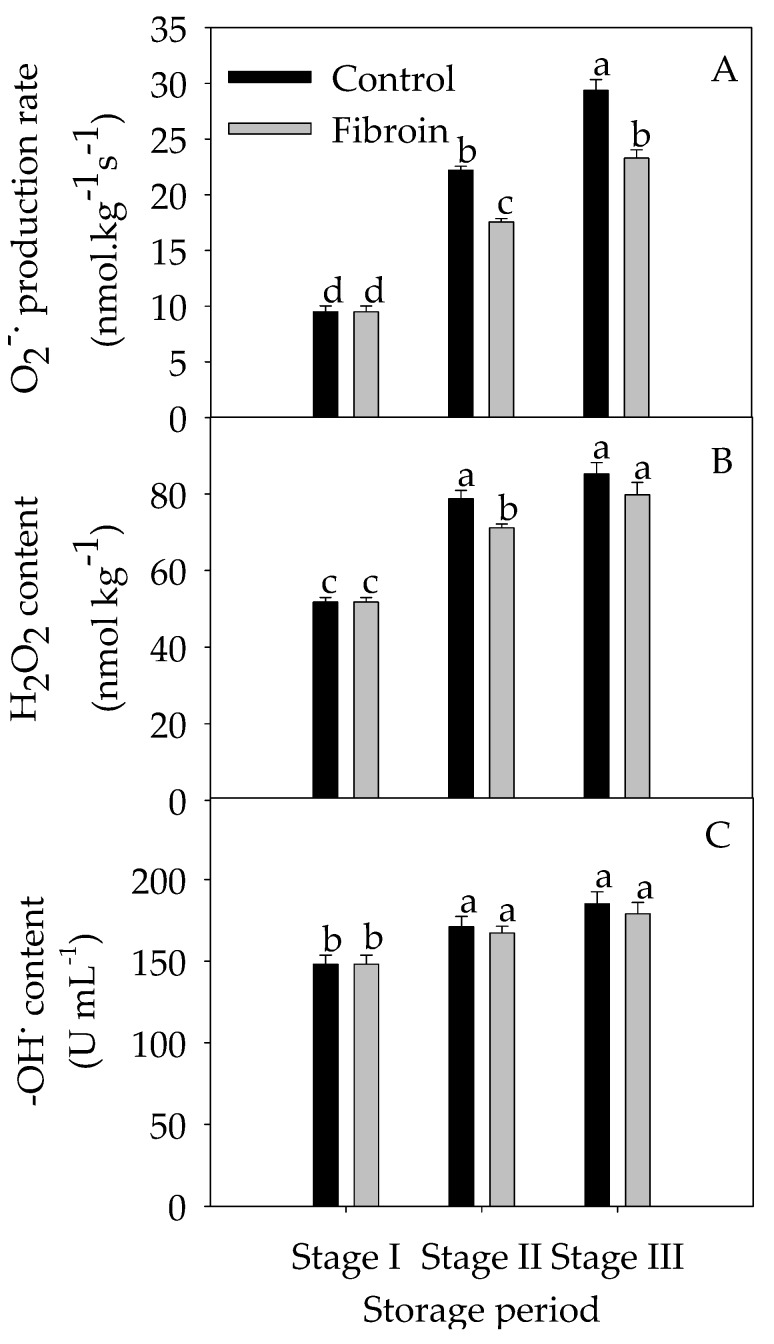
Effects of fibroin treatment on O_2_^−.^ production rate (**A**) and contents of H_2_O_2_ (**B**) and −OH^.^ (**C**) in peels of harvested banana fruit during cold storage. The data are presented as means ± standard errors (*n =* 3). The means with different letters are significantly different (*p* < 0.05).

**Figure 5 metabolites-09-00152-f005:**
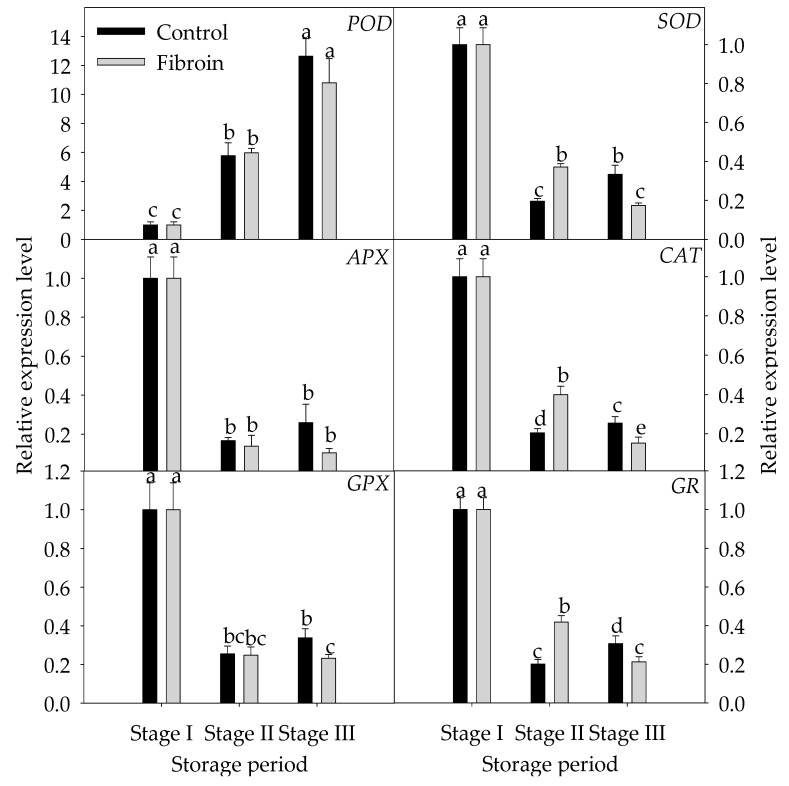
Relative gene expression levels of *POD*, *SOD*, *APX, CAT, GPX,* and *GR* in peels of harvested banana fruit during cold storage. The data are presented as means ± standard errors (n = 3). The means with different letters are significantly different (*p* < 0.05).

**Figure 6 metabolites-09-00152-f006:**
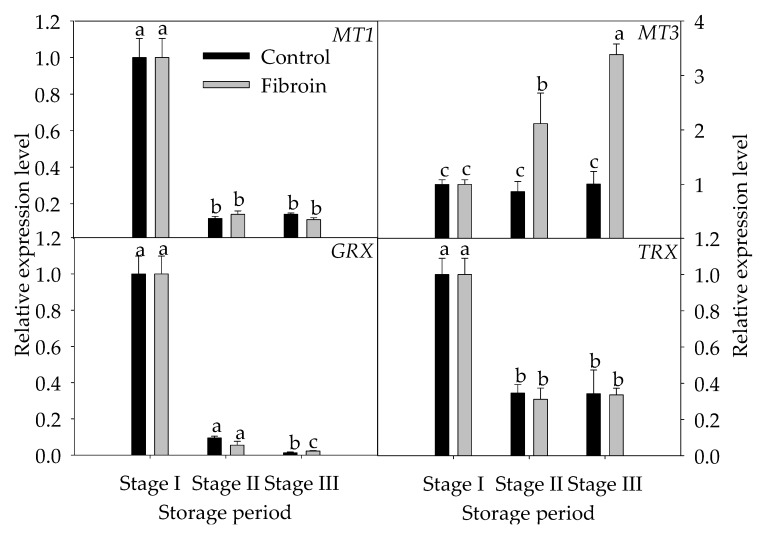
Relative gene expression levels of *MT1, MT2, GRX,* and *TRX* in peels of harvested banana fruit during cold storage. The data are presented as means ± standard errors (*n =* 3). The means with different letters are significantly different (*p* < 0.05).

**Figure 7 metabolites-09-00152-f007:**
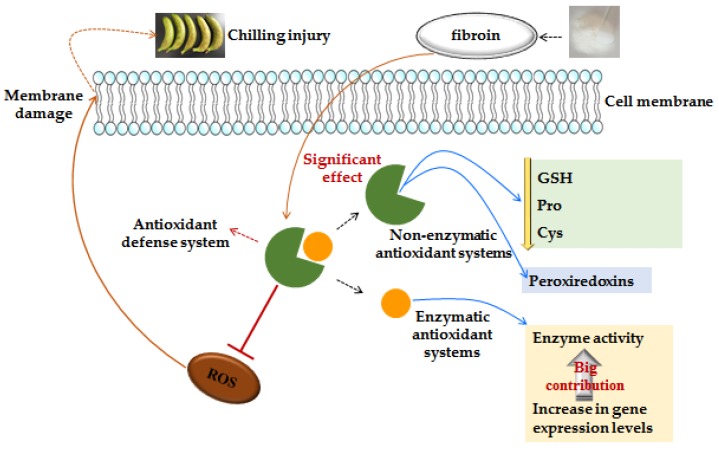
The possible mechanism of peel browning inhibition via enhanced antioxidant capability of harvested banana peel by fibroin treatment.

**Table 1 metabolites-09-00152-t001:** Contents of major amino acids in harvested banana fruit during storage. The data are presented as means ± standard errors (*n* = 3). The means with different letters are significantly different (*p* < 0.05).

Amino Acid Name	Treatment	Amino Acid Content (mg kg^−1^)		
		Stage I	Stage II	Stage III
Asp	Control	95.7 ± 5.0 ^a^	105.2 ± 4.2 ^b^	105 ± 2.7 ^b^
	Fibroin	95.7 ± 5.0 ^a^	116.8 ± 6.3 ^b^	125.5 ± 3.1 ^c^
Thr	Control	10.1 ± 0.2 ^a^	8.1 ± 0.7 ^bc^	7.2 ± 0.8 ^c^
	Fibroin	10.1 ± 0.2 ^a^	8.9 ± 0.3 ^b^	9.5 ± 1.0 ^ab^
Ser	Control	37.2 ± 0.5 ^a^	35.4 ± 2.1 ^a^	31.3 ± 1.5 ^b^
	Fibroin	37.2 ± 0.5 ^a^	36 ± 1.8 ^a^	34 ± 2.6 ^ab^
Asn	Control	145.6 ± 2.5 ^a^	135.2 ± 5.6 ^b^	106.3 ± 3.9 ^e^
	Fibroin	145.6 ± 2.5 ^a^	159.4 ± 6.8 ^c^	189.5 ± 2.4 ^d^
Glu	Control	51.3 ± 4.3 ^a^	19.3 ± 1.2 ^b^	19.6 ± 2.5 ^b^
	Fibroin	51.3 ± 4.3 ^a^	25.4 ± 2.3 ^c^	20.9 ± 1.1 ^d^
Gly	Control	18.4 ± 1.6 ^a^	15.3 ± 1.8 ^a^	3.4 ± 0.6 ^b^
	Fibroin	18.4 ± 1.6 ^a^	3.6 ± 0.5 ^b^	3.7 ± 0.7 ^b^
Ala	Control	22.5 ± 2.0 ^a^	37.5 ± 1.8 ^b^	3.2 ± 0.1 ^c^
	Fibroin	22.5 ± 2.0 ^a^	37.9 ± 2.6^ b^	40.9 ± 4.6 ^b^
Val	Control	3.1 ± 0.2 ^a^	3.2 ± 0.1 ^a^	1.3 ± 0.1 ^b^
	Fibroin	3.1 ± 0.2 ^a^	3.6 ± 0.3 ^a^	3.8 ± 0.2 ^a^
Cys	Control	0.0 ± 0.0 ^a^	0.0 ± 0.0 ^a^	0.0 ± 0.0 ^a^
	Fibroin	0.0 ± 0.0^ a^	2.1 ± 0.1^ b^	4.3 ± 0.3 ^c^
Met	Control	2.3 ± 0.1^ a^	1.2 ± 0.2^ b^	3.4 ± 0.4 ^c^
	Fibroin	2.3 ± 0.1^ a^	1.3 ± 0.1^ b^	3.7 ± 0.1 ^c^
P-Ser	Control	10.6 ± 0.3^ a^	10.2 ± 1.5^ a^	10.1 ± 0.6^ a^
	Fibroin	10.6 ± 0.3^ a^	10.3 ± 2.4^ a^	11.5 ± 0.3 ^b^
Ile	Control	2.8 ± 0.1^ a^	3.5 ± 0.2 ^b^	4.3 ± 0.1 ^c^
	Fibroin	2.8 ± 0.1^ a^	3.7 ± 0.3 ^b^	2.6 ± 0.2^ a^
Tyr	Control	2.0 ± 0.1^ a^	3.1 ± 0.3 ^bc^	2.8 ± 0.5 ^bc^
	Fibroin	2.0 ± 0.1^ a^	2.7 ± 0.2 ^b^	3.4 ± 0.3 ^c^
Phe	Control	2.8 ± 0.1^ a^	2.2 ± 0.3 ^b^	1.3 ± 0.1 ^c^
	Fibroin	2.8 ± 0.1^ a^	1.5 ± 0.1 ^c^	2.9 ± 0.2^ a^
β-Ala	Control	3.6 ± 0.3^ a^	3.2 ± 0.2^ a^	3.1 ± 0.3^ a^
	Fibroin	3.6 ± 0.3^ a^	3.2 ± 0.3^ a^	5.3 ± 0.6^b^
P-Eta	Control	3.3 ± 0.2^ a^	2.1 ± 0.2^b^	3.0 ± 0.1^ a^
	Fibroin	3.3 ± 0.2^ a^	3.9 ± 0.3 ^c^	3.2 ± 0.3^ a^
GABA	Control	7.2 ± 0.5^ a^	5.6 ± 0.4 ^b^	6.8 ± 0.4^ ab^
	Fibroin	7.2 ± 0.5^ a^	6.4 ± 0.7^ ab^	9.7 ± 0.2 ^c^
His	Control	67 ± 5.3^ a^	58 ± 4.1 ^b^	50 ± 3.7 ^c^
	Fibroin	67 ± 5.3^ a^	60 ± 3.6^ ab^	67 ± 2.9^ a^
Orn	Control	3.9 ± 0.5^ a^	2.2 ± 0.2 ^b^	2.0 ± 0.3 ^b^
	Fibroin	3.9 ± 0.5^ a^	2.2 ± 0.1 ^b^	2.1 ± 0.1 ^b^
Lys	Control	4.6 ± 0.2^ a^	0.0 ± 0.0 ^b^	0.0 ± 0.0 ^b^
	Fibroin	4.6 ± 0.4^ a^	1.0 ± 0.1 ^c^	2.7 ± 0.3 ^d^
Arg	Control	6.5 ± 0.6^ a^	5.4 ± 0.3 ^b^	3.3 ± 0.7 ^d^
	Fibroin	6.5 ± 0.6^ a^	13.2 ± 0.6 ^c^	6.4 ± 0.2^ a^
Pro	Control	0.0 ± 0.0^ a^	2.3 ± 0.1 ^b^	2.5 ± 0.1 ^b^
	Fibroin	0.0 ± 0.0^ a^	4.7 ± 0.3 ^c^	7.3 ± 0.3 ^d^

**Table 2 metabolites-09-00152-t002:** The content of primary metabolites in peels of harvested banana fruit during cold storage. The data are presented as means ± standard errors (*n* = 3). The means with different letters are significantly different (*p* < 0.05).

Name	Primary Metabolite Content (µg Ribitol g^−1^ Fresh Weight (FW))					
	Stage I		Stage II		Stage III	
	Control	Fibroin	Control	Fibroin	Control	Fibroin
**Amino Acids**						
l-Norvaline	3.3 ± 0.7^ a^	3.3 ± 0.7^ a^	1.9 ± 0.1 ^bc^	1.6 ± 0.2 ^c^	1.5 ± 0.07 ^d^	2.1 ± 0.1 ^b^
l-Serine	11.3 ± 2.4^ a^	11.3 ± 2.4^ a^	7.2 ± 0.3 ^b^	6.8 ± 0.7 ^b^	5.9 ± 0.7 ^b^	10.8 ± 1.3 ^c^
l-Threonine	4.8 ± 0.3^ a^	4.8 ± 0.3^ a^	2.3 ± 0.1 ^b^	1.9 ± 0.3 ^b^	1.8 ± 0.2 ^b^	2.1 ± 0.5 ^b^
Glycine	3.3 ± 0.5 ^c^	3.3 ± 0.5 ^c^	11.3 ± 0.1^ a^	9.5 ± 0.4 ^b^	9.4 ± 0.2 ^b^	10.8 ± 1.3^ a^
l-Proline	62.5 ± 9.4^ a^	62.5 ± 9.4^ a^	12.1 ± 0.2 ^b^	9.7 ± 0.5 ^b^	9.7 ± 0.3 ^b^	10.0 ± 0.5 ^b^
l-Aspartic acid	58.5 ± 17.8^ a^	58.5 ± 17.8^ a^	32.2 ± 1.0 ^c^	26.4 ± 1.2 ^d^	22.3 ± 1.0 ^e^	34.6 ± 2.2 ^b^
l-Asparagine	1.8 ± 0.6 ^e^	1.8 ± 0.6 ^e^	7.3 ± 0.1 ^b^	5.9 ± 0.1 ^c^	3.6 ± 0.2 ^d^	9.3 ± 0.2^ a^
l-Valine	106.0 ± 17.8^ a^	106.0 ± 17.8^ a^	3.8 ± 0.2 ^d^	5.0 ± 0.1 ^c^	3.4 ± 0.2 ^d^	6.4 ± 2.2 ^b^
**Sugars**						
Glucopyranose	17.3 ± 4.7^ a^	17.3 ± 4.7^ a^	11.4 ± 0.7^ a^	5.8 ± 1.7 ^c^	8.2 ± 0.3 ^c^	14.8 ± 1.0 ^b^
α-d-Glucopyranoside	1510.0 ± 334.1^ a^	1510.0 ± 334.1^ a^	786.2 ± 14.8 ^b^	660.5 ± 66.2 ^c^	801.8 ± 25.9 ^b^	817.9 ± 30.3 ^b^
2-Keto-d-gluconic acid	11.7 ± 0.9^ a^	11.7 ± 0.9^ a^	4.8 ± 0.5 ^c^	2.6 ± 0.8 ^d^	3.4 ± 0.2 ^d^	6.1 ± 0.4 ^b^
dtABLE-Glucuronic acid	0.9 ± 0.6 ^d^	0.9 ± 0.6 ^d^	4.2 ± 0.2^ a^	2.2 ± 0.6 ^c^	1.8 ± 0.2 ^c^	3.5 ± 0.3 ^b^
**Organic acids**						
Propanoic acid	77.1 ± 5.3^ a^	77.1 ± 5.3^ a^	7.1 ± 0.7 ^b^	4.7 ± 0.8 ^b^	6.5 ± 0.6 ^b^	4.9 ± 2.2 ^b^
Citric acid	125.7 ± 12.0^ a^	125.7 ± 12.0^ a^	99.3 ± 1.1 ^b^	86.7 ± 12.0 ^b^	71.8 ± 0.8 ^c^	125.2 ± 3.7 ^d^
Malic acid	108.4 ± 26.2^ a^	108.4 ± 26.2^ a^	73.9 ± 2.3 ^c^	62.1 ± 7.6 ^c^	73.6 ± 1.8 ^c^	95.3 ± 3.1 ^b^
6-Hydroxy-2-aminohexanoic acid	31.8 ± 1.7^ a^	31.8 ± 1.7^ a^	11.5 ± 0.3 ^b^	8.3 ± 1.6 ^c^	5.6 ± 0.4 ^d^	9.6 ± 1.2 ^c^
3,5-Dimethoxymandelic acid	16.9 ± 0.8^ a^	16.9 ± 0.8^ a^	4.3 ± 0.1 ^b^	5.7 ± 1.5 ^b^	4.2 ± 0.5 ^b^	3.6 ± 0.7 ^b^
Ethanedioic acid	1101.4 ± 260.7^ a^	1101.4 ± 260.7^ a^	641.6 ± 13.7 ^c^	625.5 ± 57.5 ^c^	503.0 ± 19.4 ^d^	740.9 ± 20.0 ^b^
**Lipids**						
Hexadecanoic acid	98.4 ± 3.3^ a^	98.4 ± 3.3^ a^	41.6 ± 6.6 ^b^	41.8 ± 11.4 ^bc^	57.8 ± 4.0 ^c^	29.2 ± 3.8 ^d^
Octadecanoic acid	60.8 ± 18.9^ a^	60.8 ± 18.9^ a^	19.7 ± 3.6 ^b^	22.2 ± 6.8 ^bc^	30.7 ± 3.0 ^c^	14.0 ± 1.2 ^d^
**Alcohols**						
2,5-Monoformal-l-rhamnitol	3.2 ± 0.7 ^d^	3.2 ± 0.7 ^d^	5.3 ± 0.09 ^b^	4.6 ± 0.4 ^c^	6.3 ± 0.4 ^a^	5.5 ± 0.2 ^b^
3,5-Dimethoxymandelic acid	16.9 ± 0.9 ^d^	16.9 ± 0.9 ^d^	4.3 ± 0.1 ^b^	5.7 ± 1.5 ^c^	4.4 ± 0.5 ^a^	3.6 ± 0.7 ^c^
**Others**						
Acetamide	48.1 ± 2.4 ^a^	48.1 ± 2.4 ^a^	29.4 ± 0.5 ^c^	30.8 ± 1.2 ^c^	43.0 ± 3.2 ^b^	30.9 ± 0.4 ^c^
Monoethanolamine	10.0 ± 1.6 ^a^	10.0 ± 1.6 ^a^	5.0 ± 0.05 ^d^	6.5 ± 0.4 ^b^	6.9 ± 0.5 ^b^	5.4 ± 0.3 ^c^
Tetradecane	0.9 ± 0.2 ^d^	0.9 ± 0.2 ^d^	5.3 ± 0.05 ^b^	6.7 ± 1.5 ^a^	7.0 ± 1.4 ^a^	5.1 ± 0.2 ^b^

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
