# Peer review of "Fibroin Delays Chilling Injury of Postharvest Banana Fruit via Enhanced Antioxidant Capability during Cold Storage"

_metabolites, 2019, doi:10.3390/metabo9070152_

Reviewer 1 Report

The paper presented by Liu et al. is interesting, reports new data on the role of fibroin treatment in delaying peel browning during postharvest of banana fruit, and should be published in Metabolites after major revision.

Comments for the author:

Regarding to this work I have some comments and minor remarks:

The authors examined significant differences by using ANOVA; however, they did not consider this analysis for the interpretation/discussion of the results:

Line 100: ”Fibroin treatment lowered the contents of citric acid, malic acid, ethanedioic acid and 6-hydroxy-2-aminohexanoic acid at the middle of storage but increased them significantly by the end of storage”. According to results and statistical analysis fibroin treatment did not lower the contents of citric acid, malic acid and ethanedioic acid at the middle of storage with respect to control, so this sentence needs to be revised. The same needs to be considered for:

Line 103: "Fibroin treatment increased their contents initially"

Line 113: "By contrast, fibroin treatment increased the contents of GSH and ASA, but decreased the levels of total carotenoids and total phenols at the middle of storage". ASA and total phenols did not increase and decrease in response to fibroin treatment at the middle of storage, respectively.  

Line 117: According to ANOVA, POD and CAT activities were not affected in treated fruits.

Lines 144-151: please revised qRT-PCR results.

Abstract, Discussion, Conclusions and Figure 7 also need to be revised.

 For q-RT-PCR experiments (Figures 5 and 6), did the authors check the expression of other gene family members? In the cases of SOD, APX and CAT could be important to explain the discrepancies observed between gene expression and enzyme activity.

 Line 320: as metabolites were quantified, please indicate the use of calibration curves.  

 Line 138: "levels of genes"

 Line 217: "production"

 Line 233: "system"

 Line 251: "α-carotene and β-carotene"

 Line 278: "It was shown that MT gene promptly up-regulated under oxidative stress and oxidative stress stimulated the transcription of MT gene [39]". Confused sentence, please clarify.

Reviewer 2 Report

Dear Drs.

In the attachment you will find two notes.

line 55, indicates that it is convenient to increase more information about fibroin and its role in postharvest.

In line 296, it would be convenient to indicate what harvest index you have used. As well as if the fibroin solution was purchased (indicate company, in this case) or was manufactured from silk cocoons.

On the other hand, please review especially the paragraphs of the material and methods section in order to reduce the similarity index.

Thanks

Author Response

Round  2

Reviewer 1 Report

The authors have addressed all my concerns and the paper should be published in Metabolites after minor revision.

Comments for the authors:

Line 286: "As shown in Figure 6, fibroin treatment up-regulated the transcription level of GRX significantly by the end of storage but down-regulated the transcription level of GRX slightly" should be "As shown in Figure 6, fibroin treatment up-regulated the transcription level of GRX significantly by the end of storage"

Line 337: "The results of all primary metabolites were expressed as µg g-1 of fruit fresh weight", this is not correct. If the authors have not used calibration curves, then metabolite determinations should be expressed as relative levels: metabolite quantification is based on the relative peak response area of each chromatogram and expressed relative to the internal standard (ribitol) and the sample fresh weight.  Absolute quantification (µg g-1) can be obtained if calibration curves of specific metabolites are included in the GC-MS run. Please also correct Table 2, as you have relative levels to internal standard (ribitol) and not absolute levels.
